medicinal chemistry

cognitive impairments, oxidative stress, Morris water maze, lead, bi-enzymes

**Authors for correspondence:**

Yanguang Yang
e-mail: yyg193@sina.com
Haiying Gu
e-mail: hygu@ntu.edu.cn
Donglin Xia
e-mail: xiadonglin@ntu.edu.cn

This article has been edited by the Royal Society of Chemistry, including the commissioning, peer review process and editorial aspects up to the point of acceptance.

†These authors contributed equally to this work.

# Bi-enzymes treatments attenuate cognitive impairment associated with oxidative damage of heavy metals

Chao Chen[1,†], Xiaoxin Zhang[3,†], Hao Huang[1], Hongyi Bao[1], Xiaodong Li[1], Ye Cheng[2], Jing Zhang[1], Yin Ding[4], Yanguang Yang[5], Haiying Gu[1] and Donglin Xia[1]

[1]School of Public Health, and [2]Xinglin College, Nantong University, Nantong, Jiangsu 226019, People's Republic of China
[3]Boao Evergrande International Hospital, Qionghai, Hainan 571400, People's Republic of China
[4]State Key Laboratory of Analytical Chemistry for Life Science, School of Chemistry and Chemical Engineering, Nanjing University, Nanjing, Jiangsu 210093, People's Republic of China
[5]Nantong Tumor Hospital, Nantong, Jiangsu 226361, People's Republic of China

DX, 0000-0003-2819-9384

Oxidative stress has been implicated in the pathogenesis of cognitive impairment. Lead (Pb) is a common environmental toxicant and plays a vital role in oxidative stress activation. In this study, a superoxide dismutase (SOD) and catalase (CAT) containing poly (lactic-co-glycolic acid) (PLGA) meso-particles (PLGA@SOD-CAT) were prepared to attenuate cognitive impairment via inhibiting oxidative stress in rats. It was prepared using a double emulsion (water/oil/water phase) technique to minimize the hazardous effects of Pb burden on cognitive impairment. The meso-particles antagonized the Pb-induced cognitive impairments. Behaviour, serum biochemical parameters and biomarkers of oxidative stress in rats were evaluated after they were subjected to intravenous injection with lead nitrate and PLGA@SOD-CAT. Moreover, the potential protective mechanism of PLGA@SOD-CAT was determined. Notably, PLGA@SOD-CAT appreciably agented memory impairment caused by lead nitrate and it could significantly inhibit Pb-induced oxidative stress in the blood. Furthermore, a remarkable reversion effect of cognitive impairments, including escape latency, crossing platform

times and time per cent during the platform quadrant, after PLGA@SOD-CAT administration were noted. Therefore, these results suggested that the bi-enzymes platform was a superior product in eliminating Pb-induced cognitive impairments through reducing expression of Pb-associated oxidative stress, and it could potentially be applied in detoxifying heavy metals in blood circulation.

## 1. Introduction

Heavy metal, lead (Pb), is a highly toxic heavy metal and is considered a harmful residue to human beings and other living creatures [1,2]. Pb has been in use for making several types of products. Toxic lead compounds have remarkably been released from the earth's crust into the atmosphere, water, soil and living organisms [3–7]. Of note, plants easily absorb and accumulate Pb in different tissues, this endangers human health through the food chain or inhalation [8]. Its deposition is evident in several tissues such as the liver, kidney, bones and brain [9]. Therefore, Pb is considered a public health concern.

Besides, Pb is known as a 'chemical time bomb' due to its toxicity which causes adverse effects on nervous and haemopoietic systems [10]. Children are particularly vulnerable to the neurotoxic effects of Pb exposure due to their developmental state, body weight and behaviours that increase the risk of exposure. Notably, about 45% of Pb is absorbed from the gastrointestinal tract in children whereas adults absorb between 10% and 15% under continuous exposure. Pb thereafter circulates in the blood and accumulates in bones, and can access the brain of children resulting in cognitive impairments. In children with higher inhalation rates per unit of body, weaker Pb detoxification capabilities exhibit high blood lead levels for a long duration [11,12]. Similar effects have been confirmed in cases of Pb exposure via inhalation pathways [13]. Consequently, there is a need to conduct in-depth research on the development of the nervous system as it is the primary site for adverse effects of environmental Pb exposure [14–16].

The blood–brain barrier effectively protects the nervous system against heavy metals; however, Pb causes brain damage manifested mainly by neurological system dysfunction [17,18]. Cerebral injury caused by Pb can lead to serious complications, such as excess deposition *in vivo*, significant reduction activates of the acetylcholine neurotransmitter. Moreover, Pb stimulates the formation of free radicals and reactive oxygen species (ROS). The inflammatory cytokines transmitted by blood have attracted different levels of research [19,20]. Pb enters the gastrointestinal tract through the intake of foods and water, it is absorbed and permeates into the blood. Furthermore, the deposit of Pb in bones is released to the blood continuously, which would cause poisoning. Previous studies have reported that the haematologic system is an important target for lead-induced toxicity. Therefore, we postulated that brain injury results from oxidative stress in the blood due to accumulated Pb, as it cannot easily be transferred through the blood–brain barriers [21].

Increasing studies have suggested that Pb potentially causes oxidative damage which initiates inflammatory reactions by stimulating the formation of free radicals and reactive oxygen species (ROS) [20]. It can be worsened when the activity of superoxide dismutase (SOD) and catalase (CAT) are depressed. Excess Pb level in the blood alters the activation of SOD resulting in ROS accumulation. Increased oxidative stress has been associated with reduced systemic and cerebral antioxidant status [22]. Current reports have proved that increased oxidative stress causes behavioural and cognitive deficits, yet interventions that mitigate oxidative stress should attenuate/overcome neurobehavioural deficits [23–25]. The inducers of SOD and CAT scavenge for ROS during blood circulation. This study, therefore, hypothesized that a long-circulating bi-enzymes meso-sphere system consisting of SOD and CAT (PLGA@SOD-CAT) could provide sustained ROS catalysis. Extensive researches have been conducted to functionalize the PLGA surface in enhancing the performance of the PLGA-based drug delivery system [26–28]. The poly (lactic-co-glycolic acid) (PLGA) was selected as the SOD-CAT carrier with prolonged *in vivo* circulation time. It catalysed the conversion of hydrogen peroxide ($H_2O_2$) to water, thereby potentially reduced cognitive impairment and neuronal damage. Collectively, findings from this study suggested that the bi-enzyme meso-sphere system can potentially be applied in detoxifying heavy metals in blood circulation.

## 2. Material and methods

### 2.1. Materials and animals

Superoxide dismutase (SOD) and catalase (CAT) were obtained from Aladdin Biochemical Technology Co. Ltd (Shanghai, China). mPEG and PLGA (Mn = 15 000) were obtained from Sigma-Aldrich

Co. Ltd. All other reagents were analytical grade and were used without further purification/modification.

Specific pathogen-free male Sprague Dawley (SD) rats ($180 \pm 10$ g), obtained from the Experimental Animal Center, Nantong University (Nantong China) were used in the experiments. Five rats were housed in each cage within a conventional animal facility ($22 \pm 3°C$, 45–70% relative humidity; 12 h light/dark cycle) and were fed with standard pelleted food and water ad libitum. All procedures involving animals and their care were approved by the institutional committee of animal care.

## 2.2. PLGA@SOD-CAT synthesis

Exactly mPEG-PLGA was dissolved in 2 ml ethyl acetate (oil phase). After oscillatory dissolution, 30 mg SOD (100 mg ml$^{-1}$) was added; 30 mg CAT was dissolved in 0.3 ml $K_2HPO_4$ (0.05 mol l$^{-1}$) solution to achieve a pH of 7.4 (aqueous phase). The aqueous phase (0.3 ml) and the oil phase (2 ml) were placed in a 10 ml centrifugal tube and shocked with a vortex oscillator. Colostrum was formed using an ultrasonic crusher in an ice bath for 3 min and added to a mixed solution of 7 ml PVA solution (2 wt%) and 3 ml F68 solution (2 wt%), stirred vigorously for 15 min at room temperature. Thereafter, the solution was sonicated for 5 min in an ice bath to form a double emulsion. The solvent was rotated to evaporate at 30°C, centrifuged at 1000 r.p.m. for 10 min, washed thrice and filtered for subsequent use.

## 2.3. Characterizations

Morphological changes were assessed by scanning electron microscopy (SEM) using a JSM-6700F microscope (JEOL, Japan). The Malvern zeta sizer (ZEN3690, Malvern Instruments Ltd, UK) was used to determine zeta potential change of PLGA@SOD-CAT. The drug loading efficiency of SOD-CAT was calculated as follows:

$$\text{Drug loading (\%)} = \frac{\text{weight of loaded drug}}{\text{total weight of microspheres}} \times 100\%$$

To quantify the kinetics of vesicular release, sudden release of CAT (water-soluble) in phosphate-buffered saline (PBS) during 24 h was monitored.

The PLGA@SOD-CAT was kept in PBS or BSA solution (40 mg ml$^{-1}$) for 20 days. Particle stability was assessed by measuring their size over 20 days.

## 2.4. Measuring the reactive oxygen species

Total ROS was measured by staining the cells using the 2′,7′-dichlorofluorescin diacetate (DCFDA) cellular ROS detection assay kit (Beyotime, China). Blood samples were collected and incubated with 2,7-dichlorodihydrofluoresceine diacetate (DCFH-DA) at 37°C for 20 min. Thereafter, the cells were washed with PBS and dichlorofluorescein (DCF) fluorescence determined using a fluorescence microplate reader (Safire2 Tecan, Mannedorf, Switzerland). The excitation filter was set at 485 nm whereas the emission filter was set at 528 nm.

## 2.5. Mitochondrial membrane potential measurements

$1 \times 10^6$ epithelial cells were plated in a six-well plate in complete medium and treated with lead nitrate, lead nitrate + free SOD-CAT or lead nitrate + PLGA@SOD-CAT for 24 h. The mitochondrial transmembrane potential was determined with the 5,5′,6,6′-tetrachloro-1,1′,3,3′-tetraethyl benzimidazolyl carbocyanine iodide (JC-1) reagent (1:100). After 20 min incubation, the media were removed and washed with PBS. The fluorescence of the JC-1 monomer (green) and aggregate (red) was measured in a confocal laser scanning microscope (FV 3000, Olympus, USA) with excitation/emission settings at 514/529 nm and 585/590 nm, respectively.

## 2.6. Behavioural assessment

The 20 rats were randomly divided into four groups: (i) control group, administrated with 0.1 ml PBS via the tail vein at the first day, (ii) Pb ion-treated group, administrated with 6.5 mg kg$^{-1}$ d$^{-1}$ lead nitrate for 5 days, (iii) free SOD-CAT group, administrated with 6.5 mg kg$^{-1}$ d$^{-1}$ lead nitrate for 5 days, 1.99 µg SOD and 2.47 µg CAT via the tail vein at the first day, and (iv) PLGA@SOD-CAT group, administrated with

$6.5 \, \text{mg} \, \text{kg}^{-1} \, \text{d}^{-1}$ lead nitrate for 5 days and 0.1 ml PLGA@SOD-CAT (with SOD $19.87 \, \mu\text{g} \, \text{ml}^{-1}$, CAT $24.67 \, \mu\text{g} \, \text{ml}^{-1}$) via the tail vein at the first day.

The water maze consisted of an off-white circular pool (80 cm in diameter) with the upper part surrounded by a 40 cm-high Perspex wall and filled with water at $25 \pm 1^\circ\text{C}$ [29]. The circular pool was segmented into four quadrants and enclosed in a curtain. The motions caused by rats were recorded by a camera. The training procedure included four trials each day with four different starting positions from 16th to 20th day. In each trial, rats were allowed to find the escape platform from the centre of the southwest quadrant of the pool within 90 s. All rats were allowed to rest on the platform for 15–20 s after the platform was mounted. Thereafter, the rats were led to the platform and kept for 15 s. The hidden platform task (spatial reference memory) was performed on the 21st day. The escape latency, escape length, swimming speed and rounds at which the rats crossed the platform was recorded.

Besides, body weight, the ratio of brain/body weight, lead concentration in the blood and brain of different groups were obtained after 0, 4, 8, 12, 16 and 20 days of the experiment.

## 2.7. Blood routine and biochemical measurements

In this experiment, the rats were sacrificed and whole blood was collected. The red blood cells (RBC), white blood cells (WBC), platelets (PLT) and haemoglobin (HGB) were detected using the animal haematology analyser (950FS, Drew Scientific, USA). Serum samples were obtained by centrifuging 2 ml of whole blood at 2500 r.p.m. for 15 min. Creatinine, urea nitrogen, glutamic-pyruvic and glutamic oxalacetic transaminase were detected by the automatic biochemical analyser (TRI-9002, Trilogy, USA).

## 2.8. Detecting inflammatory factors, neurotransmitters and oxidative indicators

The rats were sacrificed after the Morris water maze test and the hippocampal and brain tissues were obtained. The factors including creatinine, malondialdehyde (MDA), interleukine-6 (IL-6), hydrogen peroxide ($H_2O_2$) and tumour necrosis factor (TNF), were assayed using the ELISA kit. The superoxide dismutase (SOD), $H_2O_2$ and MDA were selected as oxidation indexes and their levels were determined using the equivalent assay kits. Of note, MDA content was determined using the thiobarbituric acid method and SOD activity was tested using the xanthine oxidase method. The enzymatic activity was evaluated by $H_2O_2$ decomposition measured at 405 nm.

## 2.9. Statistical analysis

Statistical analysis was performed using SPSS 19.0. The results were expressed as mean ± standard deviation, and means between the two groups were compared using the Student $t$-test. One-way analysis of variance was used for statistical analysis. $p < 0.05$ was considered statistically significant.

# 3. Results and discussion

High blood lead levels increased the ROS and thus required more enzyme to eliminate [30]. The Pb was released into the blood, in the long term after the bones were enriched. PLGA meso-sphere encapsulation was applied to achieve the long-term release of SOD-CAT (figure 1*a*).

## 3.1. Preparation and characterization of PLGA@SOD-CAT

The PLGA was approved by US food and drug administration (FDA) because of excellent biocompatibility and biodegradability for efficient drug loading. The PLGA nanoparticles (PLGA@SOD-CAT) were loaded with SOD and CAT as a 'core', PLGA as the 'coat' structure. As the PLGA concentration increased, the drug (SOD and CAT) loading efficiency decreased (figure 1*b*). That is to say, the coating behaviour of PLGA became more and more complete. As confirmed by the result in figure 1*c*, the sudden release rate of CAT from PLGA@SOD-CAT decreased as the PLGA concentration increased. Ten per cent of PLGA was chosen in the synthesis of PLGA@SOD-CAT to obtain round in shape and approximately 1.1 μm in diameter particles at room temperature (figure 1*d*). Moreover, PLGA@SOD-CAT showed even dispersion and relatively uniform size based on dynamic light scattering results (figure 1*e*). The haemolytic activities of SOD, CAT and PLGA@SOD-CAT were evaluated toward rat RBCs. As shown in electronic supplementary material, figure S1, no haemolytic effect was assessed in all groups.

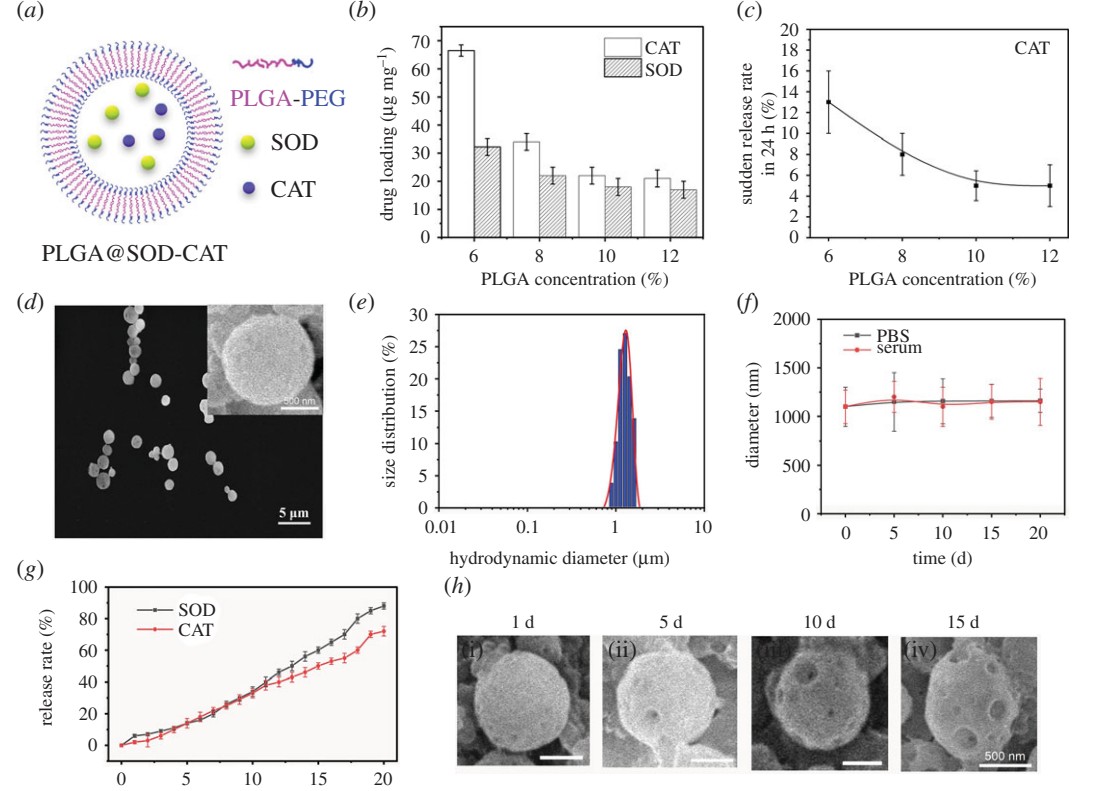

**Figure 1.** PLGA@SOD-CAT characterization results. (*a*) Schematic diagram. (*b*) Drug loading vs increasing PLGA concentration. (*c*) The sudden release rate in 24 h. (*d*) SEM images. (*e*) The particle size distribution. (*f*) Stability in PBS and serum. (*g*) *In vitro* release profile. (*h*) SEM images during the releasing process (i, 1 d; ii, 5 d; iii, 10 d; iv, 15 d).

Of note, several distinct proteins are present in blood circulation. However, there was a need to identify the proteins that highly interacted with PLGA. Therefore, a control experiment was conducted to determine whether this interaction increased when PLGA@SOD-CAT was faced with the proteins. Bovine serum albumin (BSA, 40 g l$^{-1}$) was selected as a competitive protein to examine the stability of PLGA@SOD-CAT. The PLGA@SOD-CAT was kept in PBS with 40 mg ml$^{-1}$ of BSA for 20 days. The amount of PLGA@SOD-CAT in the supernatant was measured as shown in figure 1*e*. The amount of PLGA@SOD-CAT in the supernatant did not significantly increase in the presence of BSA ($p > 0.05$) compared with the PLGA@SOD-CAT group incubated with pure PBS. This partly illustrated that PLGA potentially loaded the bi-enzymes (SOD and CAT) stably in the presence of several competitive proteins. Nanoparticles are known to be eliminated by phagocytes such as macrophages whereas macroparticles are not conducive for long circulation in the blood. Notably, the PLGA@SOD-CAT prepared in this study accommodated long blood circulation and reduced cell phagocytosis.

Furthermore, the PLGA microspheres encapsulated bi-enzymes were established which gradually released SOD and CAT for approximately 20 days (figure 1*g*). Unsurprisingly, the water-soluble SOD showed a sudden release at the first 3 days, maybe due to the water-soluble particles being more permeable than oil-soluble CAT. The surfaces of PLGA@SOD-CAT were found no-hole, small-hole and large-hole surfaces as the time went on (figure 1*h*). The gradual natural release increased the concentration under *in vivo* conditions thereby reducing oxidative stress. The encapsulation efficiency of SOD was slightly higher than the CAT, possibly due to the high molecular weight. The *in vitro* cytotoxicity was evaluated (electronic supplementary material, figure S2), the PLGA@SOD-CAT exhibited slight cytotoxicity against endothelial cells.

## 3.2. Release behaviour and reduced oxidative stress

Pb has been reported to induce cognitive damage by increasing the reactive oxygen species content and depleting the antioxidant enzyme levels [31,32]. The SOD and CAT had been reported to be the key enzymes for ROS detoxification. The sustained release profile of PLGA@SOD-CAT could increase the

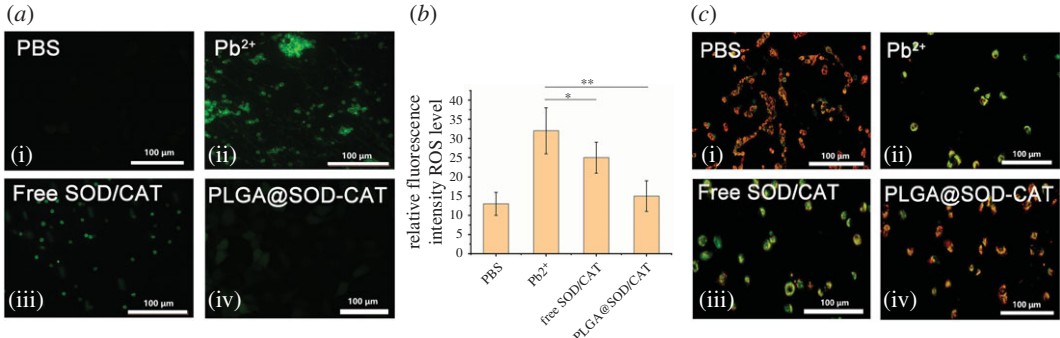

**Figure 2.** *In vitro* effect of PLGA@SOD-CAT in eliminating reactive oxygen species. (*a*) Representative images of ROS staining and (*b*) ROS fluorescence intensity in per cell. (*c*) The fluorescence value of red and green was determined by flow cytometry, after staining with JC-1. The data are presented as means ± s.d., *n* = 10. *$p < 0.05$, **$p < 0.01$. i, PBS; ii, Pb²⁺; iii, free SOD/CAT; iv, PLGA@SOD-CAT.

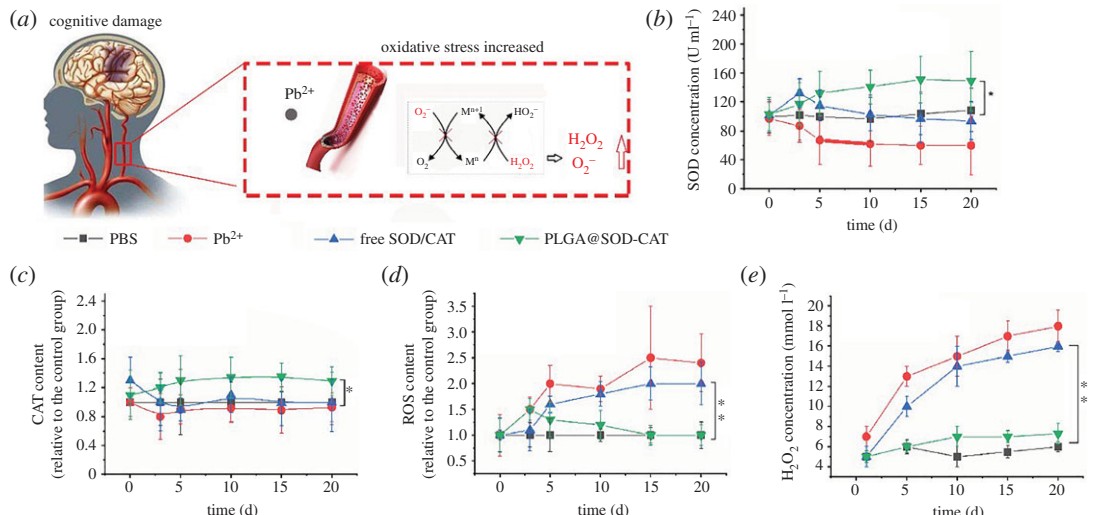

**Figure 3.** The effect of PLGA@SOD-CAT in eliminating reactive oxygen species in blood. (*a*) Schematic diagram showing Pb causes oxidative damage, which initiates inflammatory reactions by stimulating the formation of free radicals and reactive oxygen species (ROS). (*b* and *c*) SOD and CAT *in vivo* concentration, as PLGA@SOD-CAT was administered by intravenous administration. (*d*) ROS production measured as intensity of DCF fluorescence and expressed as a percentage value of the control sample. (*e*) Corresponding concentration profile for $H_2O_2$ in the different treated groups. *n* = 10. *$p < 0.05$, **$p < 0.01$.

lifespan of SOD and CAT to improve the effects of ROS detoxification, which was experimentally proved in figure 2*a*,*b*. The ROS fluorescence intensity in PLGA@SOD-CAT-treated group was significantly reduced ($p < 0.01$).

As the heavy metal would hamper the electron transfer of the mitochondrion, causing abnormal mitochondria membrane potential, the cyanine dye JC-1 was used to evaluation detoxification of PLGA@SOD-CAT. As shown in figure 2*c*, green signals from the Pb²⁺-treated group were strong, indicating low mitochondrial membrane potential (abnormal condition) and red signals from the PLGA@SOD-CAT-treated cells were found, indicating high mitochondrial membrane potential.

*In vivo*, Pb could induce cognitive damage by increasing the reactive oxygen species content, too (figure 3*a*). *In vivo* release dynamics were tested to investigate whether PLGA@SOD-CAT release behaviour reduced the oxidative stress in the blood. A decrease in SOD and CAT activity was reported in the lead ion-treated group (figure 3*b*,*c*), this inhibited the electron transport of the enzyme resulting in a concomitant decline in ROS levels and $H_2O_2$ concentration. Following the short initial burst release in the first 3 days, the free SOD-CAT could not play along with activity in scavenging effect to ROS. Of note, increasing antioxidant effect was observed in the PLGA@SOD-CAT-treated group. This resulted in a subsequent decline in ROS levels and activated the platelets for up to 20 days. Moreover, it was observed that PLGA@SOD-CAT significantly reduced the intracellular ROS levels (figure 3*d*) and $H_2O_2$ generation in the blood (figure 3*e*), ($p < 0.01$).

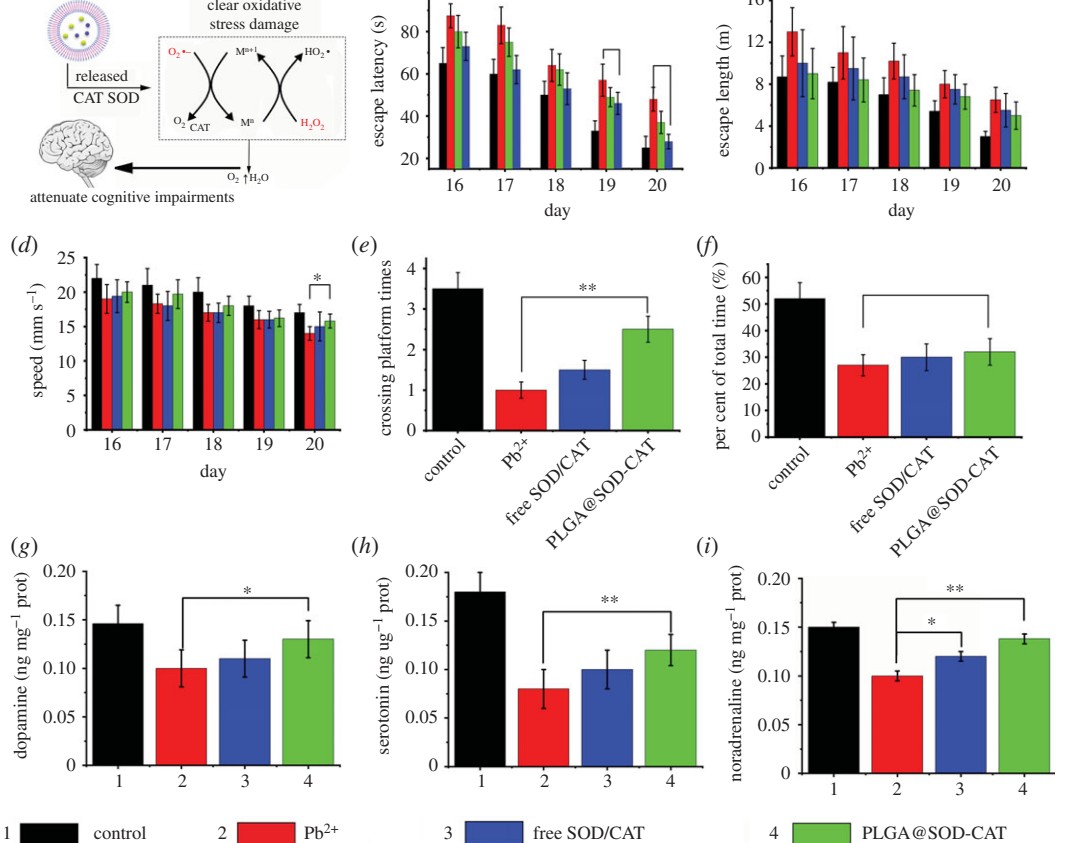

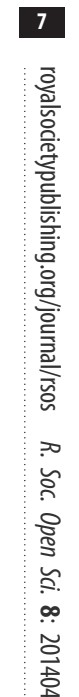

**Figure 4.** Effects of different groups on spatial learning task impairment and memory loss in young rats using the Morris water maze test. (*a*) Experimental procedure of the Morris water maze test. (*b*) The escape latencies of rats in the spatial learning task during the 4 training days. (*c*) The escape length of rats in the test. (*d*) Motion speed of rats in the probe test. (*e*) Crossing platform times of rats in the test. (*f*) Time per cent during the platform quadrant in the probe test. The change of neurotransmitters at 20th day. (*g*) Dopamine, (*h*) 5-hydroxytryptamine (5-HT) and (*i*) noradrenaline in different groups. $n = 10$ per group. $^*p < 0.05$, $^{**}p < 0.01$.

## 3.3. Morris water maze

Notably, Pb exposure would result in brain damage and prevent the growth of neurons [16]. Therefore, the Morris water maze test was used to evaluate the effects of lead-induced spatial learning task impairment and memory loss. This catalysed the conversion of $H_2O_2$ to water and thereby potentially reduced cognitive impairment and neuronal damage (figure 4*a*). The spatial learning of rats tested in the Pb ion-treated group showed the longest escape latency and length compared with the other groups (figure 4*b*,*c*). The escape latency and length were shorter in the 20th day after PLGA@SOD-CAT treatment ($p < 0.01$). There was no significant difference between the free SOD-CAT group, PLGA-treated group and the control group based on the latency time during the experiment (electronic supplementary material, figure S3).

The control group showed the fastest speed whereas the Pb ion-treated group was the slowest (figure 4*d*). The PLGA@SOD-CAT group was faster than the Pb and free SOD-CAT groups on the 20th day ($p < 0.05$). Crossing platform times and per cent of total time during the platform quadrant of four groups are shown in figure 4*e*,*f*. Moreover, in the Pb ion-treated group, the crossing platform times and per cent of total time during the platform quadrant significantly decreased as compared with the control. The PLGA@SOD-CAT group significantly improved in the crossing platform times ($p < 0.01$) and per cent of total time during the platform quadrant ($p < 0.05$) as compared with the Pb ion-treated group. These results suggested that PLGA@SOD-CAT potentially exhibited a neuroprotective effect in the Pb-treated rats.

## 3.4. The change of neurotransmitters

The effect of PLGA@SOD-CAT on neurotransmitters was examined to understand the potential regulatory functions of the significant differentially expressed neurotransmitters including dopamine (DP), 5-hydroxytryptamine (5-HT) and noradrenaline in rat brain [33]. The long-circulating bi-enzymes

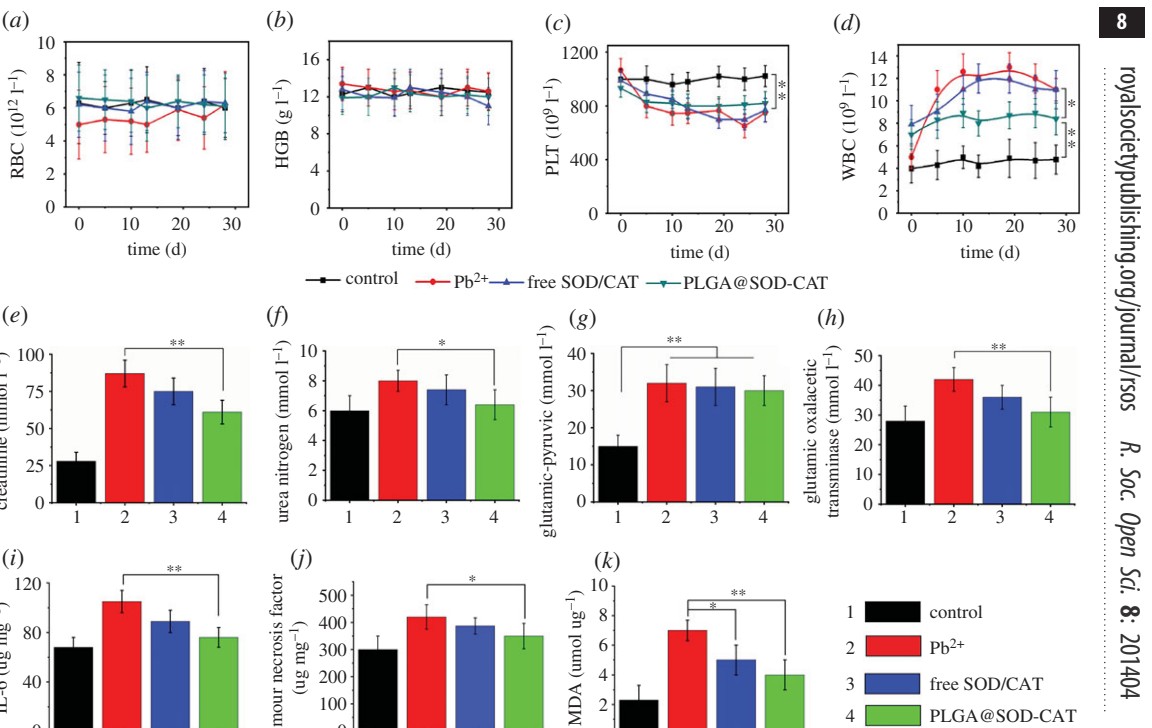

**Figure 5.** Blood routine examination, biochemical criterion and inflammatory factors of different groups. (a) RBC, (b) HGB, (c) PLT, (d) WBC, (e) creatinine, (f) urea nitrogen, (g) glutamic-pyruvic, (h) glutamic oxalacetic transaminase, (i) IL-6, (j) tumour necrosis factor, (k) MDA. The data are presented as the mean ± s.e.m., $n = 10$ per group. $^*p < 0.05$, $^{**}p < 0.01$.

meso-sphere system consisting of SOD and CAT (PLGA@SOD-CAT) could provide sustained ROS catalysis. It catalysed the conversion of $H_2O_2$ to water and thereby potentially reduced cognitive impairment and neuronal damage. In the Pb ion-treated group, the three neurotransmitters significantly decreased as compared with the control group (figure 4g–i). The transient release in the free SOD-CAT group would not improve the situation; only noradrenaline concentration slightly increased. The PLGA@SOD-CAT could significantly improve dopamine ($p < 0.05$), 5-hydroxytryptamine ($p < 0.01$) and noradrenaline ($p < 0.01$) as compared with the lead ion-treated group. Furthermore, oxidative stress reduced due to a decrease in oxidative damage. This was caused by a continuous increase in SOD and CAT levels induced by the release behaviour of PLGA@SOD-CAT. The activated neurotransmitters in rat brain could be recovered through treatment with PLGA@SOD-CAT. This concurred with the results of spatial learning ability shown in figure 4b–f.

## 3.5. PLGA@SOD-CAT effect on the blood routine and biochemical measurements

A series of routine blood examinations were performed and results are shown in figure 5a–d. The total counts of RBC and HGB were maintained at a normal range after treatment with Pb, free SOD-CAT or PLGA@SOD-CAT. Also, PLT count decreased in the other three groups except for the control group. The total WBC quantity in the blood reflects the overall body immunity and assists in detecting inflammation. Notably, the WBC count increased after Pb ion treatment. The PLGA@SOD-CAT group exhibited a significant decrease in the WBC count ($p < 0.01$). This suggested that PLGA@SOD-CAT potentially reduced inflammation. Moreover, serum biochemical values were assessed in rats. Of note, rats who received Pb ion treatment had dramatically increased biochemical levels for rat blood compared with the control group ($p < 0.01$). However, treatment with PLGA@SOD-CAT significantly decreased the creatinine ($p < 0.01$), urea nitrogen ($p < 0.05$) and glutamic oxalate levels ($p < 0.01$), as shown in figure 5e–h.

## 3.6. Detecting inflammatory factors and oxidative indicators

Interleukin-6 and tumour necrosis factor were significantly affected by the metallic lead. The highest concentrations of IL-6 and tumour necrosis factor were reported in the lead exposure group

(figure 5i–k). Besides, the concentration of IL-6 and tumour necrosis factor decreased after being treated with PLGA@SOD-CAT, suggesting that PLGA@SOD-CAT potentially reduced inflammation. The highest concentration of superoxide dismutase was reported in the lead exposure group, whereas MDA concentration was the lowest showing a statistical difference.

# 4. Conclusion

SOD and CAT are crucial enzymes involved in reducing oxidative stress. In this study, the protective effect of PLGA@SOD-CAT against Pb-induced cognitive impairments was studied. Based on the findings, PLGA@SOD-CAT showed a remarkable effect on the behavioural test, the release of bi-enzymes and reduced oxidative stress in the blood of Pb-treated rats. Notably, PLGA@SOD-CAT played a role in regulating Pb-induced disorder of the neurotransmitters in rats. This suggests that PLGA@SOD-CAT is an effective product in eliminating Pb-induced cognitive impairments and it can potentially be applied in detoxifying heavy metals in blood circulation.

Ethics. All animal procedures were approved by the Nantong University ethical committee and reported according to the ARRIVE guidelines. All animal experiments were performed according to the relevant laws and institutional guidelines inspected by the Division of Comparative Medicine at Nantong University (protocol no. 20190225-017).

Data accessibility. Our entire data are deposited at the Dryad Digital Repository: https://doi.org/10.5061/dryad.7m0cfxprv [34]. All data generated or analysed during this study are included in this article.

Authors' contributions. D.X., H.G. and Y.Y. conceived the idea and designed the experiments. D.X. and C.C. did the synthesis and characterization. D.X. and H.H. conducted the *in vitro* experiments, D.X., X.L., J.Z., Y.D. and Y.C. did the *in vivo* analyses. D.X. conducted the bioinformatics analyses. D.X., Y.D., H.G. and C.C. analysed all the data and wrote the manuscript. All the authors have proof-read this article and approved its publication.

Competing interests. We have no competing interests.

Funding. The authors thank the National Key Research and Development Program of China (grant no. 2018YFF0215500), the National Natural Science Foundation of China (grant no. 21874077), Key R&D projects of Jiangsu Province (grant no. BE 2019690), the Science and Technology R&D Fund of Nantong City (grant no. JC2019139) and Large Instruments Open Foundation of Nantong University.

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
