## [Reviewer comments · Royal Society Open Science]

Review History

RSOS-201404.R0 (Original submission)

Review form: Reviewer 1

Is the manuscript scientifically sound in its present form?

Yes

Are the interpretations and conclusions justified by the results?

Yes

Is the language acceptable?

Yes

Do you have any ethical concerns with this paper?

No

Have you any concerns about statistical analyses in this paper?

No

Recommendation?

Accept with minor revision (please list in comments)

Comments to the Author(s)

Chen et al. reported the superoxide dismutase (SOD) and catalase (CAT) containing PLGA meso-particles (PLGA@SOD-CAT) to attenuate cognitive impairment via inhibiting oxidative stress in rats. The results suggested that the bi-enzymes platform is a superior product in eliminating Pb-induced cognitive impairments through reducing expression of Pb-associated oxidative stress. The paper is well written, well-structured, and easy to follow. The data shown is informative and the mechanism was thoroughly discussed. The materials and methods section is comprehensive, although there are experimental sections that require clarification. Below are comments for the authors to consider which I think will help improve the overall presentation of the manuscript:

1. As the authors description that the PLGA@SOD-CAT can potentially be applied in detoxifying heavy metals in blood circulation in the Abstract. Could the authors please comment in this in their conclusions?
2. The PLGA@SOD-CAT was demonstrated intravenous administration in Figure 3,4, and 5. Cytotoxicity studies, hemolysis tests should be performed and shown in the text.
3. In figure 1G, there was a sudden release of SOD in the initial, but not happen in the CAT, why? Pleas clarify in the text.
4. The PLGA@SOD-CAT could attenuate cognitive impairment via inhibiting oxidative stress. But SOD and CAT are known for the presence of extensive in vivo. Is the PLGA play an important role?
5. The test of membrane potential (JC-1) should be described in the method.

Review form: Reviewer 2

Is the manuscript scientifically sound in its present form?

Yes

Are the interpretations and conclusions justified by the results?

No

Is the language acceptable?

Yes

Do you have any ethical concerns with this paper?

No

Have you any concerns about statistical analyses in this paper?

No

Recommendation?

Major revision is needed (please make suggestions in comments)

Comments to the Author(s)

1. In the animal model, how long are lead nitrate be administering? Fives days? How long are PLGA-sod/cat be administering?
2. In fig.4, We can find that the result of PLGA-sod/cat group is better than free sod/cat group. Whether PLGA will have some effect? can you add a PLGA group?

Decision letter (RSOS-201404.R0)

Dear Professor Xia:

Title: Bi-enzymes treatments attenuate cognitive impairment associated with oxidative damage of heavy metals

Manuscript ID: RSOS-201404

The editor assigned to your manuscript has now received comments from reviewers. We would like you to revise your paper in accordance with the referee and Subject Editor suggestions which can be found below (not including confidential reports to the Editor). Please note this decision does not guarantee eventual acceptance.

Please submit your revised paper before 07-Nov-2020. Please note that the revision deadline will expire at 00.00am on this date. If we do not hear from you within this time then it will be assumed that the paper has been withdrawn. In exceptional circumstances, extensions may be possible if agreed with the Editorial Office in advance. We do not allow multiple rounds of revision so we urge you to make every effort to fully address all of the comments at this stage. If deemed necessary by the Editors, your manuscript will be sent back to one or more of the original reviewers for assessment. If the original reviewers are not available we may invite new reviewers.

RSC Associate Editor:
Comments to the Author:
(There are no comments.)

RSC Subject Editor:
Comments to the Author:
(There are no comments.)

Reviewers' Comments to Author:
Reviewer: 1

Comments to the Author(s)

Chen et al. reported the superoxide dismutase (SOD) and catalase (CAT) containing PLGA meso-particles (PLGA@SOD-CAT) to attenuate cognitive impairment via inhibiting oxidative stress in rats. The results suggested that the bi-enzymes platform is a superior product in eliminating Pb-induced cognitive impairments through reducing expression of Pb-associated oxidative stress. The paper is well written, well-structured, and easy to follow. The data shown is informative and the mechanism was thoroughly discussed. The materials and methods section is comprehensive, although there are experimental sections that require clarification. Below are comments for the authors to consider which I think will help improve the overall presentation of the manuscript:

1. As the authors description that the PLGA@SOD-CAT can potentially be applied in detoxifying heavy metals in blood circulation in the Abstract. Could the authors please comment in this in their conclusions?
2. The PLGA@SOD-CAT was demonstrated intravenous administration in Figure 3,4, and 5. Cytotoxicity studies, hemolysis tests should be performed and shown in the text.
3. In figure 1G, there was a sudden release of SOD in the initial, but not happen in the CAT, why? Pleas clarify in the text.
4. The PLGA@SOD-CAT could attenuate cognitive impairment via inhibiting oxidative stress. But SOD and CAT are known for the presence of extensive in vivo. Is the PLGA play an important role?
5. The test of membrane potential (JC-1) should be described in the method.

Reviewer: 2

Comments to the Author(s)

1. In the animal model, how long are lead nitrate be administering? Five days? How long are PLGA-sod/cat be administering?
2. In fig. 4, we can find that the result of PLGA-sod/cat group is better than free sod/cat group. Whether PLGA will have some effect? can you add a PLGA group?

Author's Response to Decision Letter for (RSOS-201404.R0)

See Appendix A.

Decision letter (RSOS-201404.R1)

Dear Professor Xia:

Title: Bi-enzymes treatments attenuate cognitive impairment associated with oxidative damage of heavy metals
Manuscript ID: RSOS-201404.R1

It is a pleasure to accept your manuscript in its current form for publication in Royal Society Open Science. The chemistry content of Royal Society Open Science is published in collaboration with the Royal Society of Chemistry.

RSC Associate Editor
Comments to the Author:
(There are no comments.)

Reviewer(s)' Comments to Author:

Appendix A

Responses to reviewer's comments

Re: RSOS-201404

Dr Laura Smith

Publishing Editor, Journals

Dear Dr Laura Smith,

We hope you are keeping well at this difficult and unusual time!

Many thanks for your e-mail dated 15-Oct-2020. We have revised our manuscript by taking into account all the suggestions and comments from the two reviewers. Furthermore, we also followed the editorial suggestion and checked the whole manuscript. All changes are highlighted with a yellow pen in the review-only copy of the revised manuscript.

We hope this revised revision will be acceptable for publication on RSOS. Here we would like to specifically address the concerns from the editor and reviewers.

Sincerely yours,

Donglin Xia

Point-by-point reply to the reviewers.

Reviewer #1:

Chen et al. reported the superoxide dismutase (SOD) and catalase (CAT) containing PLGA meso-particles (PLGA@SOD-CAT) to attenuate cognitive impairment via inhibiting oxidative stress in rats. The results suggested that the bi-enzymes platform is a superior product in eliminating Pb-induced cognitive impairments through reducing expression of Pb-associated oxidative stress. The paper is well written, well-structured, and easy to follow. The data shown is informative and the mechanism was thoroughly discussed. The materials and methods section is comprehensive, although there are experimental sections that require clarification. Below are comments for the authors to consider which I think will help improve the overall presentation of the manuscript:

We thank the reviewers for the overall appreciation of our work and his/her concerns are addressed point-by-point as following:

Q1. As the authors description that the PLGA@SOD-CAT can potentially be applied in detoxifying heavy metals in blood circulation in the Abstract. Could the authors please comment in this in their conclusions?

A1: Thank you. According to the reviewer's suggestion, the detail is now added into the conclusion (**Page 5**), and reads:

“This suggests that PLGA@SOD-CAT is an effective product in eliminating Pb-induced cognitive impairments and it can potentially be applied in detoxifying heavy metals in blood circulation.”

Q2. The PLGA@SOD-CAT was demonstrated intravenous administration in Figure 3,4, and 5. Cytotoxicity studies, hemolysis tests should be performed and shown in the text.

A2. We thank the reviewer for raising such important issue.

According to the reviewer's suggestion, cytotoxicity studies, hemolysis tests were

characterized and shown in Supporting Figure S1 and S2. The detail is now added into the main text (Page 4), and reads:

“The hemolytic activities of SOD, CAT and PLGA@SOD-CAT were evaluated toward rat RBCs. As shown in Supporting Figure 1, no hemolytic effect was assessed in all groups.”

“The in vitro cytotoxicity was evaluated (Supporting Figure 2), the PLGA@SOD-CAT exhibited slight cytotoxicity against endothelial cells.”

Figure S1. Hemolytic activity of SOD (19.87 $\mu\text{g}/\text{mL}$), CAT (24.67 $\mu\text{g}/\text{mL}$), and PLGA@SOD-CAT (0.1 mL) on rat erythrocytes after 1 h of incubation at 37°C (n=5). The hemolytic activity was evaluated by the spectrophotometric determination of hemoglobin released from erythrocytes. PBS (0% hemolysis) and distilled water (100% hemolysis) were used as controls. Hemolysis values $\leq 5\%$ (dashed line) is considered to be non-hemolytic.

Figure S2. Cytotoxicity of PLGA@SOD-CAT against the endothelial cell line after incubating for 24 h (n = 5). PLGA@SOD-CAT did not exhibit a significant cytotoxic effect toward endothelial cells at any of the tested concentrations.

Q3. In figure 1G, there was a sudden release of SOD in the initial, but not happen in the CAT, why? Pleas clarify in the text.

A3. We thank the reviewer for raising such important issue. The detail of the reason of sudden release of SOD in the initial is now added into the text (Page 4), and reads:

“Unsurprisingly, the water-soluble SOD shown a sudden release at the first 3 days, maybe due to the water-soluble particles with more permeable than oil-soluble CAT.”

Q4. The PLGA@SOD-CAT could attenuate cognitive impairment via inhibiting oxidative stress. But SOD and CAT are known for the presence of extensive in vivo. Is the PLGA play an important role?

A4. We thank the reviewer for raising such important issue. Aliphatic biodegradable polyester, poly (lactic acid-co-glycolic acid) (PLGA), is widely used to form drug carrier in various clinic applications because it has excellent biocompatibility and biodegradability. In this study, the PLGA did not exhibit a neuroprotective effect in the Pb treated rats from the Morris water maze test results in Supporting Figure 3.

Figure S3. Effects of different groups on spatial learning task impairment and memory loss in young rats using the Morris water maze test. (A) The escape latencies of rats in the spatial learning task during the four training days. (B) Motion speed of rats in the probe test. (C) Crossing platform times of rats in the test. (D) Time percent during the platform quadrant in the probe test. * $P < 0.05$, ** $P < 0.01$.

Q5. The test of membrane potential (JC-1) should be described in the method.

A5. Thank you for your suggestions. According to the reviewer's suggestion, the test of membrane potential (JC-1) are now revised into the method (Page 3), and reads:

"1 × 10⁶ epithelial cells were plated in a 6-well plate in complete medium and treated with lead nitrate, lead nitrate + free SOD/CAT, or lead nitrate + PLGA@SOD-CAT for 24 h. The mitochondrial transmembrane potential was determined with the 5,5',6,6'-tetrachloro-1,1',3,3'-tetraethyl benzimidazolyl carbocyanine iodide (JC-1) reagent (1:100). After 20 min incubation, the media was removed and washed with PBS."

Reviewer #2.

Comments to the Author(s)

Q1. In the animal model, how long are lead nitrate be administrating? Five days? How long are PLGA-sod/cat be administrating?

A1: Thank you. We feel sorry for not explain it clearly. The detail of administration schedules is now modified in the experimental section (Page 3), and reads:

“The 20 rats were randomly divided into four groups: (1) Control group, administrated with 0.1 mL PBS via the tail vein at the first day. (2) Pb ion-treated group, administrated with 6.5 mg/kg/d lead nitrate for five days. (3) Free SOD/CAT group, administrated with 6.5 mg/kg/d lead nitrate for five days, 1.99 μ g SOD and 2.47 μ g CAT via the tail vein at the first day. (4) PLGA@SOD-CAT group, administrated with 6.5 mg/kg/d lead nitrate for five days and 0.1 mL PLGA@SOD-CAT (with SOD 19.87 μ g/mL, CAT 24.67 μ g/mL) via the tail vein at the first day.”

Q2. In fig.4, We can find that the result of PLGA-sod/cat group is better than free sod/cat group. Whether PLGA will have some effect? can you add a PLGA group?

A2: Thank you for your suggestion. There was no significant difference between the free Pb²⁺ group and the PLGA treated group based on the results of Morris water maze. It suggested that PLGA did not exhibit a neuroprotective effect in the Pb treated rats. The Morris water maze test results including the PLGA treated group is now added into the text, and reads:

“There was no significant difference between the free SOD/CAT group, PLGA treated group and the control group based on the latency time during the experiment (Supporting Figure 3).”

Figure S3. Effects of different groups on spatial learning task impairment and memory loss in young rats using the Morris water maze test. (A) The escape latencies of rats in the spatial learning task during the four training days. (B) Motion speed of rats in the probe test. (C) Crossing platform times of rats in the test. (D) Time percent during the platform quadrant in the probe test. * $P < 0.05$, ** $P < 0.01$.